# A Narrative Review of Motor Competence in Children and Adolescents: What We Know and What We Need to Find Out

**DOI:** 10.3390/ijerph18010018

**Published:** 2020-12-22

**Authors:** Luís Lopes, Rute Santos, Manuel Coelho-e-Silva, Catherine Draper, Jorge Mota, Boris Jidovtseff, Cain Clark, Mirko Schmidt, Philip Morgan, Michael Duncan, Wesley O’Brien, Peter Bentsen, Eva D’Hondt, Suzanne Houwen, Gareth Stratton, Kristine De Martelaer, Claude Scheuer, Christian Herrmann, António García-Hermoso, Robinson Ramírez-Vélez, António Palmeira, Erin Gerlach, Rafaela Rosário, Johann Issartel, Irene Esteban-Cornejo, Jonatan Ruiz, Sanne Veldman, Zhiguang Zhang, Dario Colella, Susana Póvoas, Pamela Haibach-Beach, João Pereira, Bronagh McGrane, João Saraiva, Viviene Temple, Pedro Silva, Erik Sigmund, Eduarda Sousa-Sá, Manolis Adamakis, Carla Moreira, Till Utesch, Larissa True, Peggy Cheung, Jaime Carcamo-Oyarzun, Sophia Charitou, Palma Chillón, Claudio Robazza, Ana Silva, Danilo Silva, Rodrigo Lima, Isabel Mourão-Carvalhal, Zeinab Khodaverdi, Marcela Zequinão, Beatriz Pereira, António Prista, César Agostinis-Sobrinho

**Affiliations:** 1Research Centre in Physical Activity, Health and Leisure, Faculty of Sport, University of Porto, 4200-450 Porto, Portugal; rsantos.ciafel@fade.up.pt (R.S.); jmota@fade.up.pt (J.M.); perrinha@gmail.com (P.S.); emdsr885@uowmail.edu.au (E.S.-S.); carla_m_moreira@sapo.pt (C.M.); cesaragostinis@hotmail.com (C.A.-S.); 2National Program for Physical Activity Promotion—Portuguese Directorate-General of Health, Portuguese Ministry of Health, 1049-005 Lisbon, Portugal; 3Faculty of Sport Sciences and Physical Education, University of Coimbra, Estádio Universitário de Coimbra, 3040-248 Coimbra, Portugal; mjcesilva@hotmail.com (M.C.-e.-S.); pereira.joao.rafael@gmail.com (J.P.); 4South African Medical Research Council/Wits Developmental Pathways for Health Research Unit, University of the Witwatersrand, Chris Hani Baragwanath Hospital, Chris Hani Road, Soweto, 2050 Johannesburg, South Africa; Catherine.Draper@wits.ac.za; 5Department of Sport and Rehabilitation Sciences, Research Unit on Childhood, University of Liège, Allée des sports 2, 4000 Liège, Belgium; b.jidovtseff@uliege.be; 6Centre for Sport, Exercise and Life Sciences, Coventry University, Coventry CV1 5FB, UK; ad0183@coventry.ac.uk (C.C.); aa8396@coventry.ac.uk (M.D.); 7Institute of Sport Science, University of Bern, Bremgartenstrasse 145, 3012 Bern, Switzerland; mirko.schmidt@ispw.unibe.ch; 8Priority Research Centre for Physical Activity & Nutrition, Faculty of Education & Arts, University of Newcastle, Callaghan, NSW 2308, Australia; philip.morgan@newcastle.edu.au; 9School of Education, Sports Studies and Physical Education Programme, 2 Lucan Place, Western Road, University College Cork, T12 KX72 Cork, Ireland; wesley.obrien@ucc.ie (W.O.); manosadam@phed.uoa.gr (M.A.); 10Center for Clinical Research and Prevention, Bispebjerg and Frederiksberg Hospital, Nordre Fasanvej 57, DK-2000 Frederiksberg, Denmark; peter.bentsen@regionh.dk; 11Center for Outdoor Recreation and Education, University of Copenhagen, Nødebovej 77A, DK-3480 Fredensborg, Denmark; 12Department of Movement and Sport Sciences, Vrije Universiteit Brussel (VUB), Pleinlaan 2, 1050 Brussel, Belgium; eva.dhondt@vub.be (E.D.); kdmartel@vub.be (K.D.M.); 13Inclusive and Special Needs Education Unit, Faculty of Behavioural and Social Sciences, University of Groningen, Grote Rozenstraat 38, 9712 TJ Groningen, The Netherlands; s.houwen@rug.nl; 14Research Centre in Applied Sports, Technology, Exercise and Medicine, College of Engineering, Swansea University, Wales SA1 8EN, UK; g.stratton@swansea.ac.uk; 15Department of Education and Social Work, Campus Belval, Institute for Teaching and Learning, University of Luxembourg, Porte des Sciences 11, L-4366 Esch-sur-Alzette, Luxembourg; claude.scheuer@uni.lu; 16Department for Movement and Sports, Zürich University of Teacher Education, Lagerstrasse 2, LAC—H073 CH-8090 Zürich, Switzerland; christian.herrmann@phzh.ch; 17Navarrabiomed, Complejo Hospitalario de Navarra (CHN), Universidad Pública de Navarra (UPNA), IdiSNA, 31008 Pamplona, Spain; antonio.garciah@unavarra.es; 18Laboratorio de Ciencias de la Actividad Física, el Deporte y la Salud, Facultad de Ciencias Médicas, Universidad de Santiago de Chile, USACH, 71783-5 Santiago, Chile; 19Department of Health Sciences, Navarrabiomed, CIBER of Frailty and Healthy Aging (CIBERFES), Instituto de Salud Carlos III, IdiSNA, Public University of Navarra, 31006 Pamplona, Spain; robin640@hotmail.com; 20CIDEFES, Universidade Lusófona, 1749-024 Lisbon, Portugal; antonio.palmeira@ulusofona.pt; 21Educational Sciences, University of Potsdam, Karl-Liebknecht-Str. 24/25, D-14476 Potsdam, Germany; erin.gerlach@uni-potsdam.de; 22Campus de Gualtar Edifício 4, School of Nursing, University of Minho, 4710-057 Braga, Portugal; rrosario@ese.uminho.pt; 23School of Health and Human Performance, Multisensory Motor Learning Lab, Dublin City University, 9 Dublin, Ireland; johann.issartel@dcu.ie; 24PROFITH “PROmoting FITness and Health through Physical Activity” Research Group, Department of Physical Education and Sports, Faculty of Sport Sciences, University of Granada, Ctra. Alfacar, s/n, 18017 Granada, Spain; ireneesteban@ugr.es (I.E.-C.); pchillon@ugr.es (P.C.); 25PROmoting FITness and Health through Physical Activity Research Group (PROFITH), Department of Physical and Sports Education, School of Sports Science, Sport and Health University Research Institute (iMUDS), University of Granada, Ctra. Alfacar, s/n, 18017 Granada, Spain; jruiz@ugr.es; 26Amsterdam University Medical Center, Department of Public and Occupational Health, Amsterdam Public Health Research Institute, Amsterdam UMC, Vrije Universiteit Amsterdam, 1081 HVAmsterdam, The Netherlands; s.veldman1@amsterdamumc.nl; 27Faculty of Kinesiology, Sport and Recreation, University of Alberta, Edmonton, AB T6G 2H9, Canada; zhiguan1@ualberta.ca; 28Early Start, University of Wollongong, Northfields Ave, Wollongong, NSW 2522, Australia; 29Department of Clinical and Experimental Medicine, University of Foggia, Virgilio Street, 71100 Foggia, Italy; dario.colella@unifg.it; 30Research Center in Sports Sciences, Health Sciences and Human Development, CIDESD, University Institute of Maia, ISMAI, 4475-690 Maia, Portugal; spovoas@ismai.pt; 31Department of Sports Science and Clinical Biomechanics, SDU Sport and Health Sciences Cluster (SHSC), University of Southern Denmark, 5230 Odense, Denmark; 32Sport Studies & Physical Education Department, Kinesiology, College at Brockport, State University of New York, NY 14420, USA; pbeach@brockport.edu; 33St Patrick’s Campus, DCU Institute of Education, School of Arts Education & Movement, Drumcondra, 9 Dublin, Ireland; bronagh.mcgrane@dcu.ie; 34Centro de Investigação em Educação, Campus de Gualtar, Instituto de Educação da Universidade do Minho, 4710-057 Braga, Portugal; joaosantos.iec.uminho@gmail.com; 35School of Exercise Science, Physical and Health Education, University of Victoria, Victoria, BC V8P 5C2, Canada; vtemple@uvic.ca; 36Faculty of Physical Culture, Palacky University Olomouc, třída Míru 117, 771 11 Olomouc, Czech Republic; erik.sigmund@upol.cz; 37Department of Sport Psychology, Institute of Sport and Exercise Sciences, University of Münster, Horstmarer Landweg 62b, 48149 Münster, Germany; till.utesch@uni-muenster.de; 38Department of Kinesiology and Dance, New Mexico State University, 1572 Stewart St., Las Cruces, NM 88003, USA; ltrue@nmsu.edu; 39Department of Health and Physical Education, The Education University of Hong Kong, 10 Lo Ping Road, Tai Po, Hong Kong, China; cheungpy@eduhk.hk; 40Departament of Physical Education, Faculty of Education, Social Science & Humanities, Universidad de La Frontera, Av. Francisco Salazar, 01145 Temuco, Chile; jaime.carcamo@ufrontera.cl; 41Laboratory of Adapted Physical Activity/Developmental and Physical Disabilities, School of Physical Education and Sport Science, National and Kapodistrian University of Athens, 71 Ethikis Antistasis str., 17238 Athens, Greece; sofhar@phed.uoa.gr; 42BIND-Behavioral Imaging and Neural Dynamics Center, Department of Medicine and Aging Sciences, “G. d’Annunzio” University of Chieti-Pescara, Via dei Vestini, 31, 66100 Chieti, Italy; c.robazza@unich.it; 43Research Centre on Child Studies (CIEC), Campus de Gualtar, Institute of Education, University of Minho, 4710-057 Braga, Portugal; anasilva0883@gmail.com (A.S.); beatriz@ie.uminho.pt (B.P.); 44Department of Physical Education, Campus “Prof. Aloísio de Campos”, Federal University of Sergipe, Jardim Rosa Elze, São Cristóvão 49100000, Sergipe, Brazil; danilorpsilva@gmail.com; 45Institute of Sport Science, University of Graz, 8010 Graz, Austria; rodrigoantlima@gmail.com; 46Research Center in Sports Sciences, Health Sciences and Human Development (CIDESD), University Trás-os-Montes e Alto Douro, 5001-801 Vila Real, Portugal; mimc@utad.pt; 47Department of Motor Behavior, Faculty of Physical Education and Sport Sciences, Kharazmi University, 15719-14911 Tehran, Iran; sara.khodaverdi@ymail.com; 48Center for Health Sciences and Sports, Department of Physical Education, Santa Catarina State University, R. Pascoal Simone, 358, Florianópolis 88080-350, Brazil; marcelazequinao@gmail.com; 49Faculdade de Educação Física e Desporto, Universidade Pedagógica, Maputo 1106, Mozambique; aprista1@gmail.com; 50Faculty of Health Sciences, Klaipeda University, Herkaus Manto g. 84, 92294 Klaipėda, Lithuania

**Keywords:** motor development, motor coordination, fundamental movement skills, motor proficiency, physical activity

## Abstract

Lack of physical activity is a global public health problem causing not only morbidity and premature mortality, but it is also a major economic burden worldwide. One of the cornerstones of a physically active lifestyle is Motor Competence (MC). MC is a complex biocultural attribute and therefore, its study requires a multi-sectoral, multi-, inter- and transdisciplinary approach. MC is a growing area of research, especially in children and adolescents due to its positive association with a plethora of health and developmental outcomes. Many questions, however, remain to be answered in this field of research, with regard to: (i) Health and Developmental-related Associations of MC; (ii) Assessment of MC; (iii) Prevalence and Trends of MC; (iv) Correlates and Determinants of MC; (v) MC Interventions, and (vi) Translating MC Research into Practice and Policy. This paper presents a narrative review of the literature, summarizing current knowledge, identifying key research gaps and presenting questions for future investigation on MC in children and adolescents. This is a collaborative effort from the International Motor Competence Network (IMCNetwork) a network of academics and researchers aiming to promote international collaborative research and knowledge translation in the expansive field of MC. The knowledge and deliverables generated by addressing and answering the aforementioned research questions on MC presented in this review have the potential to shape the ways in which researchers and practitioners promote MC and physical activity in children and adolescents across the world.

## 1. Introduction

The ability to move is an essential aspect of human life and development and has important implications for cognitive and social-emotional domains [1]. By inference, it is not surprising that movement is crucial for survival and also to improve and maintain quality of life [2]. Habitual physical activity comprises a wide range of movements considered vital to independence and interaction with the environment, including personal safety, functionality, leisure, performance, and well-being [3]. However, it is required that these movements are adaptive and goal-directed, goal-achieving actions [4]. The acquisition and refinement of proficiency in movement activities, involving interactions between the neuromuscular system and the environment, is commonly referred to in the literature as motor competence (MC) [5]. The term MC has also been referred as the individual’s ability to execute different motor actions, including coordination of both fine and gross motor skills that are necessary to manage everyday life [6]. Recently, Utesch and Bardid [7] have also proposed that MC denotes an individual’s degree of proficient performance in a broad range of motor skills as well as the underlying mechanisms including quality of movement, motor coordination and motor control. Due to the beneficial implications of MC on health and developmental outcomes, the study of MC—as a distinct topic—is a growing area of research.

It is important to note that MC is one of the various terms used in literature encompassing multiple aspects of human movement skills (i.e., motor proficiency, motor performance, fundamental movement/motor skill, motor ability, and motor coordination) [8,9]. The terminology is often used interchangeably, but this practice lacks precision; and, although there is considerable overlap between these concepts, they do not always refer to the same construct [10]. While there is an urgent need to review and update the terminology in this area of knowledge, this is beyond the scope of the current paper. Currently, there is no single universally accepted definition to describe MC; therefore, the original terms used by the different authors cited herein were maintained, respecting the study’s origin, how the movement outcomes were assessed (i.e., product- or process-oriented measurements), the age group involved, and the study’s aims. Additionally, the concept of physical literacy has also emerged as an interesting strategy to facilitate an increase in lifelong physical activity participation [11,12]. This multi-dimensional concept is commonly defined as the essential competence, confidence and knowledge to be physically active for life [13]. MC, while integrated in this physical literacy concept, is thus only one of its domains. In this paper, we acknowledge that there is clearly a “common ground” between these two concepts, in the sense that both claim to be important aspects to physical activity promotion across the lifespan and to be linked with health and developmental outcomes. However, the discussion whether MC is an independent area of research or an integrated part of physical literacy is still open but falls outside of the scope of the present review.

This collaborative paper aims to build upon the International Motor Competence Network (IMC Network—https://www.imcnetwork.org/). Established in late 2015, the IMC Network is a network of academics and researchers who aim to promote international collaborative research and knowledge translation in the field of MC. Within this context, we present a narrative review of the literature, summarizing the current knowledge, identifying research gaps and presenting key questions in need of future investigation on MC in children and adolescents.

## 2. Motor Competence in Children and Adolescents

Lack of physical activity is considered to be a serious global public health problem [14]. The physical inactivity pandemic causes not only morbidity and premature mortality, but is also a major economic burden worldwide [15,16,17]. One of the cornerstones of a physically active lifestyle is MC [18]. Gross MC, in particular, plays an important role in growth, development, and opportunities that lead to a physically active lifestyle [9,19,20].

The development of MC during childhood and adolescence is dependent upon—and influenced by—both biological (i.e., genetics, sex, and maturation) and environmental factors (e.g., gender roles, rearing style, stereotyping, experiences, opportunities to play, encouragement, demographics, and social factors) [21,22,23,24], as well as their interactions [25]. Indeed, children’s motor development is an expression of the integration of many physiological systems, including musculoskeletal, cardiorespiratory, sensory, and neurological systems [26] and their ability to interact with the environment [5]. Consequently, the study of motor development or the development of MC is a prerequisite for understanding human development throughout the lifespan [27].

Promoting physical activity across the lifespan is challenging. Early childhood is a critical time for the development of fundamental movement skills [28], which are considered the building blocks of more complex movements [29] and a key factor in the promotion of lifelong active lifestyles and health [4,9,19,20,30]. For this reason, understanding how fundamental movement skills are associated with physical activity across the lifespan may be a key factor in overall physical activity promotion for youth. Fundamental movement skills are generally classified in the literature into three overreaching constructs: (1) locomotor skills (e.g., run, hop, jump, slide, gallop, and leap); (2) object control/manipulative skills (e.g., strike, dribble, kick, throw, underarm roll, and catch); and (3) stability/non-locomotor skills (e.g., balancing, body rolling, bending, and twisting) [31]. Recently, Hulteen et al. [32] suggested that this classification may be too narrow and does not capture the full range of movement skills needed for the promotion of physical activity across the lifespan. Therefore, these authors propose including other skills such as resistance training, swimming and cycling, all of which require competency in specific coordinative movement patterns (e.g., swimming strokes, bodyweight squat, push-ups, etc.).

Development of MC during childhood has been recognized as a main factor for engaging in regular physical activity throughout life [19]. Indeed, proficiency in fundamental movement skills is a prerequisite for engagement in physical activities, including sport participation, which is partly due to improvements in self-regulatory mechanisms, including expectations of self-efficacy and intrinsic motivation [33]. Children with low motor proficiency may consequently prefer a less active lifestyle to avoid movement difficulties [34]. This lack of participation in physical activities is particularly concerning, given that physically inactive children are more likely to become physically inactive adults [35], and physically inactive parents tend to raise physically inactive children [36,37]. The costs and consequences of physical inactivity may, therefore, be passed on to subsequent generations, creating an intergenerational cycle of poor physical and mental health. Moreover, it is known that early life experiences are essential to build strong motor and neurodevelopmental trajectories [38] and to adopt habitual (preferably healthy) lifestyles. Therefore, understanding the most important gaps in the extant literature on this topic is a required first step to targeted research on MC in order to create positive trajectories of health and developmental outcomes in children.

In this context, innovative approaches and fresh thinking on how to improve physical activity levels in children and adolescents, particularly in girls and disadvantaged children, are required and calls for policies with impact on health, youth and sports [39]. The global physical inactivity crisis demands an urgent need to build global capacity, through multi-sectoral, inter-, multi-, and transdisciplinary approaches to tackle this pandemic [15,16].

### 2.1. Motor Competence and Its Health- and Developmental-Related Associations in Children and Adolescents

#### 2.1.1. What We Know

The importance of promoting MC development at young ages relies on the evidence that there are current and future benefits associated with the acquisition and maintenance of motor proficiency and adequate levels of physical activity [40]. Appropriate MC development contributes to children’s physical, mental, and social development, as well as to their health and well-being [27,41,42]. Indeed, evidence supports the positive association between MC and a range of health and developmental outcomes, such as healthy weight status [43,44,45,46,47], higher levels of self-esteem [48,49], perceived physical competence [20,50,51] cardiorespiratory fitness [52,53,54,55,56], muscular fitness [54], physical activity [34,43], decreased sedentary behavior [34,57], increased bone density [58], as well as better cognitive development, executive functions, school readiness, and academic achievement [8,9,59,60,61,62,63].

An interesting conceptual model by Stodden et al. [19] proposed that overweight and obesity trajectories may be triggered by the cumulative effects of low MC on reducing movement opportunities, physical fitness, and perceived MC during childhood. Overall, low MC may result in unsuccessful participation in movement play activities and/or organized sports during childhood, thus leading to a negative spiral of disengagement from an active lifestyle [9]. This idea is not new, as almost 40 years ago, Seefeldt [64] argued for the notion of a “proficiency barrier” and suggested that there might be a “critical threshold” of motor skill competence, above which children will be more active and will successfully apply fundamental movement skill competence, leading to lifetime physical activity engagement.

The notion of a “proficiency barrier” leads to the question of the efficacy of critical periods or windows of opportunity during which children may learn motor skills more easily [4]. It has been suggested that the hypothesized “proficiency barrier” may emerge during the transition from early into middle childhood (that is, during primary school years) when fundamental motor skills are ordinarily sufficiently developed, corresponding to the ages when participation in a variety of youth sports begins [5]. Even though it has been identified as one of the top ten research questions related to growth and maturation, with relevance to physical activity, fitness and performance [5], research has paid very little attention to this “proficiency barrier” [64,65]. This is in part due to the fact that there is a lack of longitudinal studies examining MC in children and adolescents.

The concept of a “proficiency barrier” intuitively considers biosocial determinants of motor coordination and development [66]. However, one may ask, what is an “adequate” level of MC at any given age, in a specific cultural context? Although some of the existing MC assessment tools have established age- and sex-adjusted normative reference values [67,68], these are usually from children who share the same culture across a limited geographic area [69]. As such, there is a dearth of international normative-referenced values for MC, irrespective of the measurement tool being used, as well as context- and health-related criterion-referenced values. The establishment of international normative and health-related criterion values of what constitutes “adequate” MC, would substantially advance the knowledge on the global prevalence of MC. It would also allow for the international monitoring of trends over time.

#### 2.1.2. What We Need to Find Out

Future research needs to address the cultural sensitivity of testing protocols and instruments. Indeed, many of the MC assessments tools were primarily designed to identify children with motor delays or difficulties and not to assess MC as a central part of human development. This leads to another important question: How should MC be assessed most effectively throughout childhood and adolescence, while also taking into account different environmental and cultural contexts?

### 2.2. Assessment of Motor Competence in Children and Adolescents

#### 2.2.1. What We Know

Early identification of children with low MC, and consequently early intervention, is both economically efficient and more effective in narrowing (and in some cases minimizing) problems associated with developmental delays in MC, than therapeutic interventions at an older age [70,71]. Moreover, planning, implementation, and evaluation of developmentally adequate movement programs depend on proper identification of the child’s actual level of motor development [72]. Ultimately, the identification of children who may have motor developmental delays is the first step to impede later difficulties [73]. Therefore, the main purpose of a MC assessment should not be only to measure (impaired) performance of skills, but rather the general traits underlying them [50,74].

There are many assessment tools to evaluate MC in children and adolescents [67,68]. The decision on how to assess a child’s MC should be determined by the purpose of the information needed and the age of the participants. MC assessment can be performed using either product- or process-oriented assessment tools [75].

A product-oriented assessment tool evaluates movement from a quantitative assessment approach. This is conducted in order to rate the outcome of skill execution, such as time, distance or frequencies of successful attempts, and thus provides little information with regard to how the movement was performed [74]. The result is generally compared to the performance of a normative group or to predefined problem situations; in this case, the result indicates whether a normatively defined minimally requirement MC level has been reached or not. In contrast, process-oriented assessments are concerned with how the skill is performed rather than the outcome of the skill, and movement is evaluated based on the demonstration of behavioral criteria, which provides information on how the movement is performed. Qualitative assessment thus provides information about the specific components of a motor task or skill that need to be improved and consequently repeated under specific prescription. Hence, these assessments can be undertaken in a more meaningful context than quantitative methods [76].

In order to capture the inherent advantages of both approaches, some assessment tools include both quantitative and qualitative items, such as the Movement Assessment Battery for Children Test—2nd Edition (MABC-2) [77]. The combination of approaches takes into account the more erratic and variable movement patterns of beginners, when compared to the more consistent patterns of skilled performers [76]. Another distinction between test instruments can be identified on the level of the test items, which can be either context-independent, movement-specific, or context-specific [68].

The preference for using a certain MC assessment tool also varies by geographic region, country, ethnicity, and socioeconomic status of the participants. For example, in Europe the KTK (Körperkoordination Test fur Kinder) [78,79]—a product-oriented non-sport specific gross motor coordination test battery—has been widely used. More recently, various versions of the Motorische Basiskompetenzen (MOBAKs) [80,81,82]—product-oriented test instruments for different school grade levels assessing the mastery of motor skills in specific situations—have been widely used across Europe, especially in Luxembourg and Switzerland as well as in 12 European countries participating in the Erasmus^+^ funded project BMC-EU (Basic Motor Competencies in Europe—http://mobak.info/bmc-eu/). In other countries, such as the USA and Australia, the Test of Gross Motor Development—2nd edition (TGMD-2) [83]—a process-oriented test battery of fundamental locomotor and object-control motor skills—has been frequently used. Of particular note, the research on convergent validity between the KTK and the TGMD-2 assessment tools has yielded only moderate correlations [75,84] showing that the different approaches measure somewhat different constructs of MC.

The use of several MC assessment tools by different studies and countries precludes direct comparisons across the globe. Indeed, there is no universal agreement about what might constitute a “gold standard” assessment of MC. Alternatively, the construction of international standardized field-based assessments of MC would ensure comparability between populations and over time. This new MC assessment tool should also ensure that socio-cultural diversity is respected. The way children and adolescents move and the movement skills that a given society expects from their children have a strong socio-cultural dependency, and therefore vary significantly worldwide [5].

There is evidence of internationally agreed assessment collaborations in the areas of health, physical activity, and fitness that have developed or adapted internationally comparable measurement tools. An example is the development of the International Physical Activity Questionnaire (IPAQ) in 1998 [85], which can be used for the standardized assessment of physical activity levels across countries. Another example is the ALPHA (Assessing Levels of Physical Activity) project, a European Union (EU) funded study that aimed to construct a set of instruments for assessing physical activity and physical fitness levels, as well as its underlying factors in a comparable way within the EU countries. The ALPHA field-based physical fitness battery is an effective illustration of a standardized test battery constructed based upon existing test instruments that has proven to be health-related, feasible, reliable, valid, and safe for school-aged children, which can be used for scientific research, as well as by practitioners (i.e., school teachers) [86,87]. As Bardid and colleagues (2015) pointed out, “The wide adoption of a single assessment tool to measure MC, has the potential to build a strong picture of how children are performing on an international level, rather than just on a national level. This would have many benefits in terms of understanding, on a global level, how motor competent children are and then proceeding to understand what cultural factors help to better facilitate MC” [88].

From a practitioner point of view (e.g., schoolteachers, physical educators, sports coaches, physicians, physiotherapists, etc.), any MC assessment tool should be valid, reliable, safe, preferably simple to use and interpret, and also feasible to apply in a school/sports club setting or in a clinical context [89,90]. This is a key aspect that is often overlooked in translating research into applied practice. Indeed, field-based assessments should serve both research and practitioners’ purposes, as they allow the monitoring of growth, development and progress of a particular child, group of children or population over time. From a public health and preventive point of view, early childhood education centers and schools play a central role in the provision and promotion of movement opportunities along with other health-related behaviors, as children spend a large amount of their time in these environments [91].

Further, there exists a proliferative interest in automating the assessment of MC using ubiquitous sensors. Recently, there have been advancements in technological and analytical capability, permitting more precise quantification of complex human movement behaviors [92,93]. Pervasive technologies, such as accelerometers, inertial measurement units and magnetometers, have been used with reasonable success. In the study of Barnes et al. [94], a magnetometer (which measures the direction, strength, or relative change of a magnetic field at a given location) was worn during a MC assessment. Using robust machine learning, data was subsequently processed, yielding visualizations of the relative performance in three-dimensions and a relative distance between children within the multi-dimensional scaling that could be used to create an automated sensor-based rank scoring, resulting in good agreement between observer and sensor scores. In a further example, Bisi et al. [95] employed inertial measurement units to compare computer-automated and human-observer assessment. Algorithm results showed agreement with raw scores assigned visually by a human observer with a mean percentage (on the entire group of children) that ranged from 82% to 100%. As such, it is important that technological developments are monitored for potential assessment of MC as our sophistication and analytical capability ensues.

It is evident that an international, standardized field-based assessment tool of MC would also be favorable for the longitudinal monitoring of MC, throughout childhood and adolescence, according to the principals of lifespan development. Indeed, the literature is still lacking in studies devoted to the prevalence of “adequate” MC, whilst limited information exists on the secular trends of MC.

#### 2.2.2. What We Need to Find Out

There is a need to develop an international standardized field-based MC assessment tool, for children and adolescents that respects the principals of lifespan development and that ensures socio-cultural diversity, allowing for comparability between populations, and the monitoring of MC levels over time.

### 2.3. Prevalence and Trends of Motor Competence in Children and Adolescents

#### 2.3.1. What We Know

As previously mentioned, data on MC prevalence and trends are limited. The available evidence about prevalence emanates from cross-sectional and longitudinal studies and describes children’s and adolescents’ MC levels as suboptimal, with rates ranging from 9% up to 52%, depending on the assessment tool being used, the year when the study was conducted, the age, and geographical origin of the participants [8,79,96,97,98,99,100]. Moreover, some evidence indicates that there is a secular decline in movement skills and movement patterns in school children [50,101,102,103,104,105]. For example, Prätorius and Milani [101] have shown that over a period of 30 years, the percentage of German children with low MC has increased substantially, from 16% in the original KTK validation study to a level of 38%.

One important point of note is that the majority of those studies investigating prevalence and trends have been conducted with relatively small samples, in different age ranges and employing different MC assessment tools, precluding a clear view on the "true" prevalence of low MC levels. Moreover, most studies were conducted in developed countries, where sedentary lifestyles are rather common among contemporary children and adolescents [106,107,108] and where pediatric overweight and obesity rates in children and adolescents are also high [109,110,111]. Therefore, there is not only a need for a standardized tool to assess MC, as stated above, but also for studies with nationally representative samples. Nevertheless, pooling data from existing studies across the world that assessed MC with the same assessment tools would provide, for the first time, an insight on the global prevalence and possibly secular trends of MC. There is evidence of well-succeeded international collaborations to pool data for physical activity, such as the International Children’s Accelerometer Database (ICAD) for example (https://www.mrc-epid.cam.ac.uk/research/studies/icad/). The ICAD initiative has pushed forward the field, by providing a global view on the prevalence and health-related associations of physical activity, in children and adolescents. Prospectively, the establishment of an international MC observatory, would ensure the global monitoring of children’s and adolescents’ MC levels over time, by supporting studies with representative samples and standardized test instruments.

Cross-cultural research could provide valuable insights regarding determinants (or correlates) of MC in different cultural contexts and how test batteries that measure specific motor skills, are culturally sensitive. However, studies assessing cross-cultural comparisons are also scarce [88,112,113,114]. For example, using the KTK as an assessment tool, Bardid et al. [88] reported that Belgian children scored higher on MC than Australian children, and that both sets of children scored lower than children assessed 40 years earlier. The authors putatively suggested that differences in educational policies and practices between countries and secular trends in sedentary behaviors may explain the results [88]. Data pooling would provide, for the first-time, important information on the cultural sensitivity of different test instruments, as well as an understanding of how different cultural practices serve to enhance or restrict children’s MC development.

#### 2.3.2. What We Need to Find Out

Pooled data and multi-center collaborative actions within an international observatory of MC, with sufficiently powered samples, examining global prevalence, trends and cross-cultural comparisons of MC are needed. These types of studies would signal the need for further investigation and calls for interventions in different countries and regions may gain heightened traction.

The level of MC of a given individual is determined by innumerable biological and environmental factors, and their interactions. As the lifestyles of the current generation of children and adolescents are substantially different from previous generations, there is a continuous need for the study of the correlates and determinants of MC.

### 2.4. Motor Competence and Its Correlates and Determinants in Children and Adolescents

#### 2.4.1. What We Know

When studying motor behavior, a developmental perspective is essential for a comprehensive understanding of movement and mobility [4]. Within an ecological framework of potential influencing factors that operate and interact at multiple levels (individual, social, environmental, and policy) [115], the study of the correlates and determinants of MC is paramount for capturing global similarities and individual differences in motor development [116].

As children grow and develop, the factors associated with MC change. In a systematic review [23] encompassing children aged 3 to 18 years old, age (increasing) was consistently identified as a correlate of MC; weight status (healthy), sex (male), and socio-economic status (higher) were consistent correlates for certain aspects of MC. However, most apparent in this review was that many potential MC correlates (e.g., psychosocial and environmental influences) had not yet been studied, suggesting that there is much to be elucidated until a theoretical framework is constructed, to explain the factors that influence the development of MC at a given age during childhood and adolescence.

It is well established that with increasing age, a gradual improvement in MC occurs; this improvement in MC is acknowledged as a general phenomenon during child development depending on the pre-dispositions of an individual and the accumulation of motor experiences, including both motor control and motor learning developmental processes [117,118,119].

With respect to biological sex, it is known that boys and girls go through the same sequence(s) of motor skill development. However, sex differences in MC, in parallel to familial sex-stereotypes or gender role models, can be found from toddlerhood onwards [120], with girls usually showing lower levels of global MC and object control/manipulation skills than boys [80,121,122,123,124]. Indeed, gender inequality is a key driver of negative health outcomes and girls seem to be given fewer opportunities to be physically active and to develop their MC than boys [106,125,126]. This is particularly concerning in those from low socio-economic backgrounds. Indeed, it is also well established that low socio-economic status is an important predictor of health impairment throughout the lifespan, contributing significantly to morbidity and premature mortality [127,128]. Moreover, it is known that health and health-related behaviors track from childhood and adolescence into adulthood, and that health inequalities are established early in life [129]. Within the behavioral explanations of the health-related associations of socio-economic status, low physical activity levels seem to play an important role [130,131]. Emerging evidence indicates that children from ethnic minorities, disadvantaged and poor environments show lower levels of MC [132]. Some disparities may exist regarding the opportunities for MC development for these children [121,122], which consequently, may lead to further disparities in physical activity levels throughout the lifespan, thereby perpetuating the cycle of poverty. For example, Temple et al. [132] found that object control skills mediated the relationship between neighborhood vulnerability and participation in physical activities among children in their first year of school. Concomitantly, children from more vulnerable neighborhoods began their school career with lower motor skill proficiency than children from less vulnerable neighborhoods.

Environmental factors are of utmost importance for MC development, as the specific physical and social context in which a child is reared and lives impacts his/her development [24]. Recently, Flôres et al. [133] concluded in their narrative review of various affordances for children’s motor development that an optimal home environment with household conditions (i.e., variety of play materials and adequate physical spaces), high family socioeconomic status, and living in a neighborhood with good potential for motor affordances is essential to children’s outdoor free play. This in turn may improve children’s motor capabilities at various ages. Regarding the school environment (i.e., physical education classes, recess, schoolyards, and playgrounds), Flôres et al. (2019) concluded that literature is generally focused on the importance of increasing physical activity levels. Nevertheless, a systematic review with pre-school children reported little evidence regarding associations between children’s fundamental motor skills competence and social and physical environment [134]. Data from The Skilled Kids Study, with Finish children, also reported that the time spent in a physical environment providing the affordances needed for physical activity was associated with higher level of MC [135].

#### 2.4.2. What We Need to Find Out

In an era of global societal changes, where income inequalities are rising and health inequalities are widening, especially among the youngest [136,137], there is an urgent need to understand the correlates and determinants of MC across different countries and regions, as well as various socio-economic and cultural backgrounds. Pooling existing data from different studies across the world, developing an international standardized field-based MC assessment tool and establishing a permanent international MC observatory, would enable researchers and practitioners to have a more robust, precise, and comprehensive overview of MC correlates and determinants. This will especially be useful for identification of subgroups within the population that are the most in need of intervention. These data will inform the design of effective intervention studies and policies, and test the validity of theory-driven models [138].

### 2.5. Motor Competence Interventions in Children and Adolescents

#### 2.5.1. What We Know

The early years (0–5 years) are a crucial period of life for raising physically competent children and should be viewed as a period of development when motor skills are acquired through structured and purposeful learning environments. There is a general acceptance that the proficiency level on a range of fundamental movement skills reflects to a large extent the degree of learning to which the individual has been exposed. Although rudimentary movement patterns may be naturally developed with free play, a mature form of MC is less likely to be achieved without appropriate practice, encouragement, feedback and instruction [31,139]. During early childhood, children should be encouraged to experience enriched environments, allowing them to achieve their full motor potential. For this reason, acquiring movement skills should be elected as central on appropriate movement programs [140].

Evidence suggests that both short-term (4–8 weeks) and long term (≥6 months) motor skill interventions are effective in improving fundamental movement skills, in children without disabilities, of both genders [140,141,142,143,144,145]. These interventions are especially effective when school and community-based programs are delivered by physical education specialists or highly trained classroom teachers [142]. However, published articles reporting intervention results often lack important details, such as the theoretical or pedagogical approach on which the intervention was based on, program intensity and duration, fidelity of implementation and characteristics of facilitators and participants [23,142]. Without such detailed information, it remains unclear to infer from available studies which factors should be targeted to ensure that the interventions are optimized and, whether, and for whom, targeted and tailored interventions should be developed [146]. Moreover, most of the existing intervention studies were conducted in children from developed countries [140], which may preclude generalizability of their implementation.

Furthermore, there is currently no quality assessment tool to specifically evaluate MC programs, despite the fact that complete program evaluations represent an important and desired prerequisite to continuous quality improvements [147]. Additionally, although a list of “candidate characteristics” of good practice of what a typically successful physical activity intervention is has been described [148], as well as the adaptation of the European Foundation for Quality Management Excellence Model for the evaluation of physical activity programs [147,149] and the construction of a self-assessment tool for physical activity programs in adults [150]; similar work focused on MC in children and adolescents is yet to be conducted. Nevertheless, such quality assessment tool for MC programs should consider the specificities of the countries and cultures where interventions take place.

Fundamental movement skills are optimally developed and ideally targeted during early and middle childhood, but many youth entering high school lack appropriate levels of MC [50,99]. This is an important area for future work to focus on [142]. However, intervention studies targeting adolescents’ MC are currently scarce. Although, the Y-PATH, a multi-component school-based RCT, showed that despite children not reaching mastery of fundamental movement skills by the age of 10 years and entering into adolescence lacking some fundamental movement skills, it is still possible to improve these via interventions during adolescence [151,152].

#### 2.5.2. What We Need to Find Out

Pooling international data on MC correlates and determinants, as well as bringing together a panel of international experts on MC interventions, would enable to advisement and support for the implementation of age, sex/gender, and socio-cultural-specific interventions for MC development in children and adolescents worldwide.

The majority of intervention studies have been conducted in early childhood education, preschools and primary/elementary school settings; however, other stakeholders may also play an important role in the promotion of children’s and adolescents’ MC. Efforts conducted by several public health institutions around the world to promote MC development (e.g., education and health systems, local authorities, community services, after school programs) would benefit from a greater understanding of evidence-based strategies to improve MC, as research needs to be translated for practice and policy.

### 2.6. Translating Motor Competence Research into Practice and Policy

#### 2.6.1. What We Know

One of the biggest research challenges is its subsequent translation into practice and policy. From awareness through acceptance and adoption, converting research findings into practice and policy has the potential to disrupt currently outdated practices [153]. Therefore, the development of integrated translation plans within research projects and networks has become an important task, urging researchers to ensure that the evidence of their studies reaches those who can benefit the most—the knowledge users.

MC development is recognized as an important component of the physical education curricula in many countries. According to McLennan and Thompson [154], a Quality Physical Education acts as “the foundation for lifelong engagement in physical activity and sport”, and offers children and adolescents appropriated learning experiences that help them to acquire psychomotor, social, emotional skills, as well as the cognitive understanding necessary to lead a physically active life. It is considered that early childhood education centers and schools should (i) implement planned movement programs as a strategy to promote MC development, (ii) ensure that physical education classes are delivered in a pedagogically appropriate manner, and (iii) that well-trained physical education teachers are engaged [140,142]. However, when working towards MC development, physical education classes and MC development programs should also consider children’s self-perceptions of MC, since it has been shown that the accuracy of children’s self-perceptions (i.e., alignment between actual MC and perceived MC) fosters their future physical activity [155].

Of note, a thorough review of good educational practices and policies across the world in relation to MC development is yet to be performed. The construction of a MC policy audit tool that provides a protocol and method for the detailed compilation and communication of country-level education policy responses on MC development and promotion, would significantly increase researchers’ and practitioners’ understanding of the most effective practices and policies on MC promotion around the world. This yet to be developed MC policy audit tool should be a standardized screening tool that will enable the monitoring of (changes in) policies and practices across countries and regions over time. The knowledge generated by the compilation of this information would be helpful to define guidelines and recommendations for the promotion of motor development in schools. This tool would provide an in-depth policy audit and cross-country comparison, highlighting similarities and differences in progress, challenges and accomplishments. Its results could reveal new ideas and opportunities for other countries.

The current World Health Organization physical activity guidelines for children and youth (from 5 to 18 years old) and for young children (<5 years old) focus on the health-related components of physical fitness (i.e., cardiorespiratory fitness and muscular strength) [156] and on free play [157], respectively. However, specific recommendations for developing MC are lacking. However, the international recommendation for the early years of the *Federation Internationale D’ Education Physique* (International Federation of Physical Education) [158], as well as some national physical activity guidelines, such as those for Finnish children [159], acknowledge the importance of motor skill development, showing that in some cultures, MC is getting more consideration. Indeed, failure to consider MC as a key antecedent of physical activity and positive health and developmental trajectories in children and adolescents likely results in treating the symptoms rather than the cause of physical inactivity and ill health.

#### 2.6.2. What We Need to Find Out

There is a need for a thorough review of good educational practices and policies across the world in relation to MC development, as well as a MC policy audit tool, for a better understanding of the most effective practices and policies on MC promotion around the world. Future international physical activity guidelines should include specific recommendations for developing MC, in children and adolescents.

## 3. Conclusions

Motor competence is a complex bio-cultural construct and, therefore, its study requires a multi-sectoral, multi-, inter-, and transdisciplinary approach. The knowledge and deliverables generated by addressing and answering the research questions and associated gaps herein presented have the potential to shape the way researchers and practitioners promote physical activity in children and adolescents across the world. It is strongly believed that these innovative progressions will open new horizons, generate future research questions and further opportunities for research, by (i) adding evidence on levels, determinants and health-related associations of motor competence in children and adolescents; (ii) pooling data from different studies across the world; (iii) developing a new international standardized field-based motor competence assessment tool; (iv) constructing a motor competence policy audit tool for education systems; (v) establishing a permanent International Motor Competence Observatory; (vi) developing recommendations on how to promote motor competence in children and adolescents; and (vii) translating research on motor competence into policy and practice. Failure to consider motor competence as a key antecedent of physical activity and positive health and developmental trajectories in children and adolescents likely results in treating the symptoms rather than the cause of physical inactivity and ill health.

In late 2015, an International Motor Competence Network of academics and researchers was established, and the current collaborative narrative review builds upon this network (https://www.imcnetwork.org/). The mission of the network is to promote international collaborative research and knowledge translation in the field of motor competence. This network represents an opportunity to push forward the scientific knowledge and develop future lines of research by improving our understanding of health, growth and developmental-related associations and determinants of appropriate levels of motor competence. Innovative approaches and fresh thinking on how to improve physical activity levels are necessary to give children the best start in life.

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
