# Peer review of "A Narrative Review of Motor Competence in Children and Adolescents: What We Know and What We Need to Find Out"

_ijerph, 2020, doi:10.3390/ijerph18010018_

Round 1

Reviewer 1 Report

This narrative review regarding the construct of motor competence is exceptionally well written. I commend the authors on the comprehensiveness of the literature review, as well as their identification of key areas needing further research. Physical inactivity, and its associated health risks, is an important area of research among children and adolescents. I think this narrative review provides researchers interested in motor competence/physical activity behavior among children and adolescents with a strong foundation from which to plan and develop much needed research projects, especially regarding the measurement of motor competence and interventions to improve motor competence. I have only a few minor suggestions for the authors to consider:

  • Line 162 - consider changing: "(iv) Correlates and Determinants of MC" being MC is identified as the dependent variable in the other questions.
  • Line 192 - consider deleting: "...is a currently and growing area of research." 
  • Line 196 - consider revising: "...interchangeably, but this practice lacks precision; and although..."
  • Line 226 - "...the pattern that is established..." It is unclear what is being referred to by pattern. Pattern of behavior?
  • Line 264 - Can you replace "claim" with "calls", as this makes more sense to me?
  • Line 273 - "An appropriate MC development contributes..." - Consider deleting "An" or adding the word program or intervention after development. I think this will make the sentence make better sense. 
  • Line 397 - 382: Why not paraphrase this rather than using a direct quote? If the authors think a direct quote is more impactful, then provide some context for the quote (e.g., who said it).  

Author Response

Reviewer 1

This narrative review regarding the construct of motor competence is exceptionally well written. I commend the authors on the comprehensiveness of the literature review, as well as their identification of key areas needing further research. Physical inactivity, and its associated health risks, is an important area of research among children and adolescents. I think this narrative review provides researchers interested in motor competence/physical activity behavior among children and adolescents with a strong foundation from which to plan and develop much needed research projects, especially regarding the measurement of motor competence and interventions to improve motor competence. I have only a few minor suggestions for the authors to consider:

  • Line 162 - consider changing: "(iv) Correlates and Determinants of MC" being MC is identified as the dependent variable in the other questions.
  • Line 192 - consider deleting: "...is a currently and growing area of research." 
  • Line 196 - consider revising: "...interchangeably, but this practice lacks precision; and although..."
  • Line 226 - "...the pattern that is established..." It is unclear what is being referred to by pattern. Pattern of behavior?
  • Line 264 - Can you replace "claim" with "calls", as this makes more sense to me?
  • Line 273 - "An appropriate MC development contributes..." - Consider deleting "An" or adding the word program or intervention after development. I think this will make the sentence make better sense. 
  • Line 397 - 382: Why not paraphrase this rather than using a direct quote? If the authors think a direct quote is more impactful, then provide some context for the quote (e.g., who said it).

Authors:

We thank this reviewer for the compliments and appreciate the comments and suggestions. The document was changed according to the reviewer’s suggestions.

Reviewer 2 Report

The work is a good review of the literature about the subject.

I Think it is signed by to many authors.

It could speak of two batteries of tests to assess the physical capacities of young people that provide normative guiding data for different ages that are correlated with motor competence (Fitnessgram and fitschool) and thus avoided references to its non-existence (line 300-308).

Makes some statements (593-604) without substantiating such indications and without carrying out any study to validate them.

Author Response

Reviewer 2

The work is a good review of the literature about the subject.

Authors:

We thank this reviewer for the compliment

I Think it is signed by to many authors.

Authors:

We understand this reviewer concern, but as stated in the manuscript, this is a collaborative effort of many researchers around the world that decided to come together and reflect upon the current state of the art in the field of MC. This narrative review has received many contributions from many researchers and experts in this field. All authors listed in the manuscript fulfil the authorship criteria.

It could speak of two batteries of tests to assess the physical capacities of young people that provide normative guiding data for different ages that are correlated with motor competence (Fitnessgram and fitschool) and thus avoided references to its non-existence (line 300-308).

Authors:

We appreciate this reviewer’s comments. In this paragraph we are referring to the lack of international /worldwide reference values for the existing motor competence test batteries. Our concept of motor competence is stated on the first paragraph of the introduction section. As such, we consider that physical fitness encompasses a different concept (as the ability of the body systems to work together efficiently allowing for health and to perform activities of daily living). Therefore, in this section of the manuscripts the Fitnessgram and fitschools test batteries were not included, because they refer to physical fitness. We do, however, mention the ALPHA test battery (a physical fitness test battery) as an effective illustration of a standardized test battery constructed based upon existing test instruments that has proven to be health-related, feasible, reliable, valid and safe for school-aged children.

Makes some statements (593-604) without substantiating such indications and without carrying out any study to validate them.

Authors:

We appreciate this reviewer’s comment. In this narrative review where we sought to summarize the current knowledge, identifying research gaps and presenting key questions in need of future investigation, in MC in children and adolescents. As such, in this particular section of the manuscript, about Translating Motor Competence Research into Practice and Policy, we are merely pointing out some of the existing gaps, speculating on some of the needs for future research and raising some questions that could be potentially solved if a thorough review of good educational practices and policies across the world in relation to MC development was performed and if a MC policy audit tool was constructed. 

Reviewer 3 Report

Review for the journal: Int. J. Environ. Res. Public Health 2020, 17, manuscript ijerph-1021898.

Title: A Narrative Review of Motor Competence in Children 3 and Adolescents: What we Know and What we Need to Find Out

Recommendation: minor revisions

GLOBAL FEEDBACK

  • This paper presents a narrative review of the literature, summarizing current knowledge, identifying key research gaps and presenting questions for future investigation on Motor Competence (MC) in children and adolescents. This is a collaborative effort from the International Motor Competence Network(IMCNetwork - https://www.imcnetwork.org/). The knowledge and deliverables generated by addressing and answering the aforementioned research questions on MC presented in this review have the potential to shape the ways in which researchers and practitioners promote MC and physical activity in children and adolescents across the world
  • This narrative review makes an important scientific and political contribution, for this reason I strongly recommend to publish the paper after a few changes (see below).

ABSTRACT

  • Abstract contains all important information

INTRODUCTION

  • the introduction is understandable and fetches the reader. The problem of a definition for MC is pointed out and thus also the necessary objective view of the subject area and the openness for various research approaches.

METHODS

  • Narrative review

RESULTS

  • Successful and compact representation of the findings

Minor comments:

Section 2.1 Motor Competence and its Health- and Developmental-related Associations in Children and Adolescents

  • Add in line 270 the subheading: What we know
  • Add before line 305 the subheading: what we need (to find out), because line 305-313 is a conclusion and so it becomes more comfortable for the reader to sum up all conclusion at the end at a glance.

Section 2: Assessment of Motor Competence in Children and Adolescents

  • Add in line 316 the subheading: What we know
  • This section is a bit „bulky“ for the reader. A suggestion: shorten Line 348 -360 use instead of the text a table (with two columns: 1: Assesment method 2:example for a test tool)
  • Including also APLPHA fitness test in this table (Line 375-378)
Assesment method example test tool
roduct-oriented assessment tools MOBAKs
process-oriented assessment tools ...
context-independent ...
movement-specific  
context-specific:  
  • Summarize the sections 409-413 ; 361-365 and 378-383 to a conclusion for the whole section 2.2, add the subheading (what we need (to find out)) for this
  • Only a suggestion: Perhaps you can add that there are still some country specific efforts to pool data (open access) comparable with databases in the activity research e.g. ICAD data base (International Children's Accelerometry Database (ICAD) - MRC Epidemiology Unit (cam.ac.uk). E.g. for Germany the data base for motor research data MOREdata: http://www.sport.kit.edu/more/english/index.php.

Kloe, M., Niessner, C., Woll, A., & Bös, K. (2019). Open Data im sportwissenschaftlichen Anwendungsfeld motorischer Tests (engl: Analysing the relevance and acceptance of the eResearch-infrastructure MO|RE data for motor research data). German Journal of Exercise and Sport Research, 49(4), 503-513. See: https://link.springer.com/article/10.1007/s12662-019-00620-2

Section 2.3. Prevalence and Trends of Motor Competence in Children and Adolescents

  • Line 422: I would suggest to add some more recent literature on secular trends e.g.

Eberhardt, T., Niessner, C., Oriwol, D., Buchal, L., Worth, A., & Bös, K. (2020). Secular trends in physical fitness of children and adolescents: a review of large-scale epidemiological studies published after 2006. International journal of environmental research and public health, 17(16), 5671.

and

Fühner, T.; Kliegl,R., Arntz, F., Kriemler, S. & Granacher , U. (2020) An Update on Secular Trends in Physical Fitness of Children and Adolescents from 1972 to 2015: A Systematic Review. Sports Medicine. https://doi.org/10.1007/s40279-020-01373-x

  • Add in line 416 the subheading: What we know
  • Replace section 444-450 at the end and add the subheading: what we need (to find out)

4. Motor Competence and its Correlates and Determinants in Children and Adolescents

  • Add in line 457 the subheading: What we know
  • add before lien 512 the subheading: what we need (to find out)

5. Motor Competence Interventions in Children and Adolescents

  • Add in line 521 the subheading: What we know
  • Add after line 560 the subheading: what we need (to find out)
  • Replace line 561-564 to the end of this section

Conclusion

  • Indication of the results in current state of research is comprehensible
  • 612-615 is a really important sentence place this in this paragraph in the overall conclusion

Author Response

Reviewer 3

GLOBAL FEEDBACK

  • This paper presents a narrative review of the literature, summarizing current knowledge, identifying key research gaps and presenting questions for future investigation on Motor Competence (MC) in children and adolescents. This is a collaborative effort from the International Motor Competence Network(IMCNetwork - https://www.imcnetwork.org/). The knowledge and deliverables generated by addressing and answering the aforementioned research questions on MC presented in this review have the potential to shape the ways in which researchers and practitioners promote MC and physical activity in children and adolescents across the world
  • This narrative review makes an important scientific and political contribution, for this reason I strongly recommend to publish the paper after a few changes (see below).

Authors:

We thank this reviewer for the compliment.

ABSTRACT

  • Abstract contains all important information

INTRODUCTION

  • the introduction is understandable and fetches the reader. The problem of a definition for MC is pointed out and thus also the necessary objective view of the subject area and the openness for various research approaches.

METHODS

  • Narrative review

RESULTS

  • Successful and compact representation of the findings

Authors:

We thank this reviewer for the compliments.

Minor comments:

Section 2.1 Motor Competence and its Health- and Developmental-related Associations in Children and Adolescents

  • Add in line 270 the subheading: What we know
  • Add before line 305 the subheading: what we need (to find out), because line 305-313 is a conclusion and so it becomes more comfortable for the reader to sum up all conclusion at the end at a glance.

Authors:

We thank this reviewer for the suggestion. The manuscript was amended accordingly.

Section 2: Assessment of Motor Competence in Children and Adolescents

  • Add in line 316 the subheading: What we know

Authors:

We thank this reviewer for the suggestion. The manuscript was amended accordingly.

  • This section is a bit „bulky” for the reader. A suggestion: shorten Line 348 -360 use instead of the text a table (with two columns: 1: Assesment method 2:example for a test tool)
  • Including also APLPHA fitness test in this table (Line 375-378)

Assesment method

example test tool

roduct-oriented assessment tools

MOBAKs

process-oriented assessment tools

...

context-independent

...

movement-specific

context-specific:

Authors:

We thank this reviewer for the suggestion. The text of the lines 348-360 (lines 364 to 377 in this revised version of the manuscript) provide only 3 examples of different test batteries (KTK, MOBAKs and TGMD-2) that have been used in different regions/countries, to make our point for the need of a having a global standardized test battery, that may allow cross-country comparisons and normative or health-related reference values. Therefore, we strongly believe that there is no need to include a table in this manuscript with existing test batteries. Because, in this section of the manuscript was not our intention to refer, thoroughly all existing test batteries, nor to indicate which we considered to be the “best” ones. In this section of the manuscript, we point out some examples of test batteries and indicate two well conducted systematic reviews that specifically address these issues (references # 68 and 69).

  • Summarize the sections 409-413 ; 361-365 and 378-383 to a conclusion for the whole section 2.2, add the subheading (what we need (to find out)) for this

Authors:

We thank this reviewer for the suggestion. The manuscript was amended accordingly.

  • Only a suggestion: Perhaps you can add that there are still some country specific efforts to pool data (open access) comparable with databases in the activity research e.g. ICAD data base (International Children's Accelerometry Database (ICAD) - MRC Epidemiology Unit (cam.ac.uk). E.g. for Germany the data base for motor research data MOREdata: http://www.sport.kit.edu/more/english/index.php.

Kloe, M., Niessner, C., Woll, A., & Bös, K. (2019). Open Data im sportwissenschaftlichen Anwendungsfeld motorischer Tests (engl: Analysing the relevance and acceptance of the eResearch-infrastructure MO|RE data for motor research data). German Journal of Exercise and Sport Research, 49(4), 503-513. See: https://link.springer.com/article/10.1007/s12662-019-00620-2

Authors:

We thank this reviewer for the suggestion. Indeed, there are some examples of data pooling, thank you for point this out. We added a couple of sentences on this and provided the ICAD example (please referrer to lines 460 to 467)

Section 2.3. Prevalence and Trends of Motor Competence in Children and Adolescents

  • Line 422: I would suggest to add some more recent literature on secular trends e.g.

Eberhardt, T., Niessner, C., Oriwol, D., Buchal, L., Worth, A., & Bös, K. (2020). Secular trends in physical fitness of children and adolescents: a review of large-scale epidemiological studies published after 2006. International journal of environmental research and public health, 17(16), 5671.

and

Fühner, T.; Kliegl,R., Arntz, F., Kriemler, S. & Granacher , U. (2020) An Update on Secular Trends in Physical Fitness of Children and Adolescents from 1972 to 2015: A Systematic Review. Sports Medicine. https://doi.org/10.1007/s40279-020-01373-x

Authors:

We appreciate the recommendation; the references were included in the manuscript as suggested (references # 105 and 106).

  • Add in line 416 the subheading: What we know
  • Replace section 444-450 at the end and add the subheading: what we need (to find out)

Authors:

We thank this reviewer for the suggestion. The manuscript was amended accordingly.

  1. Motor Competence and its Correlates and Determinants in Children and Adolescents
  • Add in line 457 the subheading: What we know
  • add before lien 512 the subheading: what we need (to find out)

Authors:

We thank this reviewer for the suggestion. The manuscript was amended accordingly.

  1. Motor Competence Interventions in Children and Adolescents
  • Add in line 521 the subheading: What we know
  • Add after line 560 the subheading: what we need (to find out)
  • Replace line 561-564 to the end of this section

Authors:

We thank this reviewer for the suggestion. The manuscript was amended accordingly.

Conclusion

  • Indication of the results in current state of research is comprehensible
  • 612-615 is a really important sentence, place this in this paragraph in the overall conclusion

Authors:

We thank this reviewer for the suggestion. The manuscript was amended accordingly.

Reviewer 4 Report

Thank you for letting me review this article. 

The article is interesting. Below are some suggestions.

  1. As the WHO pointed out, the top 10 countries of overweight/obesity in 2019 mostly located in Europe (except France and Italy due to the noble and beauty industries), US and New Zealand. The overweight rate in Japan's population is only 3%. However, in this article, mostly, the cited references came from Europe, Australia and the U.S.  It seems the review didn't touch the related researches developed in those normal weighted countries. Especially, the East, Africa, or south America. Therefore, I would like to suggest to add those related studies and submit it again.
  2. Please, tell us on what background you categorized the gaps.
  3. A meta-analysis may be needed.
  4. Line 371. the reference cited was in 2003 instead of 1998

Author Response

Reviewer 4

Thank you for letting me review this article. 

The article is interesting. Below are some suggestions.

  1. As the WHO pointed out, the top 10 countries of overweight/obesity in 2019 mostly located in Europe (except France and Italy due to the noble and beauty industries), US and New Zealand. The overweight rate in Japan's population is only 3%. However, in this article, mostly, the cited references came from Europe, Australia and the U.S.  It seems the review didn't touch the related researches developed in those normal weighted countries. Especially, the East, Africa, or south America. Therefore, I would like to suggest to add those related studies and submit it again.

Authors:

We appreciate this reviewer’s comments. Most of the literature in the field of motor competence comes from the US, Australia and European countries, and that is a fact, not a limitation of our paper. The authors of this manuscript are from all continents, throughout the manuscript we refer to papers that sustain and illustrate our ideas, we do not refer all the papers that have been published about a given idea/concept, as this is not a systematic review. However, some of our references are original papers and systematic reviews of papers conducted with children and adolescents from Asia, Africa and South American countries.

In the section 2.3. Prevalence and Trends of Motor Competence in Children and Adolescents (second paragraph) of the manuscript, we acknowledge that most of the research reporting prevalence and trends of MC, has been conducted in developed countries with high rates of pediatric overweight and obesity. We are not saying that there is no research in other developed countries with low rates of overweight/obesity. In this part of the manuscript the authors consider the references cited illustrate the point we are trying to make here.

  1. Please, tell us on what background you categorized the gaps.

Authors:

This manuscript presents a narrative review of the literature, summarizing the current knowledge, identifying key research gaps and presenting questions for future investigation on Motor Competence (MC) in children and adolescents.

  1. A meta-analysis may be needed.

Authors:

This manuscript is a narrative review, therefore a meta-analysis is not possible, given that meta-analyses require systematic reviews of the literature.

  1. Line 371. the reference cited was in 2003 instead of 1998

Authors:

The IPAQ questionnaire was developed in 1998, the publication of its validity and reliability results in 12-countries, was in 2003.